# Protective Effect of *Lactiplantibacillus plantarum* subsp. *plantarum* SC-5 on Dextran Sulfate Sodium—Induced Colitis in Mice

**DOI:** 10.3390/foods12040897

**Published:** 2023-02-20

**Authors:** Ruoran Shi, Fazheng Yu, Xueyu Hu, Yan Liu, Yuanyuan Jin, Honglin Ren, Shiying Lu, Jian Guo, Jiang Chang, Yansong Li, Zengshan Liu, Xiaoxu Wang, Pan Hu

**Affiliations:** 1State Key Laboratory for Zoonotic Diseases, Key Laboratory for Zoonosis Research of the Ministry of Education, Institute of Zoonosis, College of Veterinary Medicine, Jilin University, Changchun 130062, China; 2Institute of Special Animal and Plant Sciences of Chinese Academy of Agricultural Sciences, Changchun 130112, China

**Keywords:** inflammatory bowel disease, dextran sulfate sodium, *Lactiplantibacillus plantarum* subsp. *plantarum* SC-5, NF-κB, MAPK, tight junction protein

## Abstract

Inflammatory bowel disease (IBD) is a specific immune-associated intestinal disease. At present, the conventional treatment for patients is not ideal. Probiotics are widely used in the treatment of IBD patients due to their ability to restore the function of the intestinal mucosal barrier effectively and safely. *Lactiplantibacillus plantarum* subsp. *plantarum* is a kind of probiotic that exists in the intestines of hosts and is considered to have good probiotic properties. In this study, we evaluated the therapeutic effect of *Lactiplantibacillus plantarum* subsp. *plantarum* SC-5 (SC-5) on dextran sulfate sodium (DSS)-induced colitis in C57BL/6J mice. We estimated the effect of SC-5 on the clinical symptoms of mice through a body weight change, colon length, and DAI score. The inhibitory effects of SC-5 on the levels of cytokine IL-1β, IL-6, and TNF-α were determined by ELISA. The protein expression levels of NF-κB, MAPK signaling pathway, and the tight junction proteins occludin, claudin-3, and ZO-1 were verified using Western Blot and immunofluorescence. 16S rRNA was used to verify the modulatory effect of SC-5 on the structure of intestinal microbiota in DSS-induced colitis mice. The results showed that SC-5 could alleviate the clinical symptoms of DSS-induced colitis mice, and significantly reduce the expression of pro-inflammatory cytokines in the colon tissue. It also attenuated the inflammatory response by inhibiting the protein expression of NF-κB and MAPK signaling pathways. SC-5 improved the integrity of the intestinal mucosal barrier by strengthening tight junction proteins. In addition, 16S rRNA sequencing demonstrated that SC-5 was effective in restoring intestinal flora balance, as well as in increasing the relative abundance and diversity of beneficial microbiota. These results indicated that SC-5 has the potential to be developed as a new probiotic candidate that prevents or alleviates IBD.

## 1. Introduction

Inflammatory bowel disease (IBD) is a chronic intestinal inflammatory disease, which includes ulcerative colitis (UC) and Crohn’s disease (CD), that often involves the colon and rectum [1]. It is characterized by diarrhea, abdominal pain, and bloody stool [2]. At present, antibiotics and immunomodulators are the main drugs for the routine treatment of UC patients [3]. However, long-term use of these drugs leads to the imbalance of intestinal microbiota, antibiotic diarrhea, immunosuppression, and other negative clinical manifestations with a high recurrence rate in UC patients after treatment [4,5]. It is particularly significant to research a scientific and effective biological agent with small side effects for UC patients.

In recent years, probiotics have been widely used to treat and prevent gastrointestinal diseases [6,7]. It has also been applied in the treatment of UC because of the good therapeutic effect, no toxicity, and no interference for the normal microbiota structure [7]. *Lactiplantibacillus plantarum* subsp. *plantarum* is a type of probiotic that exists in the gut of healthy people under normal conditions, as well as in a variety of ecological niches such as vegetables, and fermented foods [8]. Studies have shown that *Lactiplantibacillus plantarum* subsp. *plantarum* is beneficial to the intestinal health for the host. Xia, Y. [9] found that *Lactiplanti bacillus plantarum* subsp. *plantarum* LP-Onlly can stably colonize the intestinal tract to compete with receptors and nutrition [9,10]. *Lactobacillus reuteri* S5can inhibit the growth and reproduction of harmful bacteria [11]. According to Peng Yu [12], *Lactiplantibacillus plantarum* subsp. *plantarum* L15 can restore the damaged intestinal mucosal barrier and reduce the inflammation level of various tissues and organs, which has great significance in relation to maintaining the balance of intestinal microbiota. In our study, *Lactiplantibacillus plantarum* subsp. *plantarum* SC-5 is a kind of stain exacted by Dongbei Suancai, which can modify microbial communities in fermented sauerkraut [13]. However, the protective effect of SC-5 for DSS-induced colitis in mice has not been clarified. Whether it has the potential to be developed as a biological agent suitable for oral administration in IBD patients needs to be further studied.

We utilized the change in colon length and the disease activity index (DAI) to assess the severity of UC. DAI score is a kind of common parameter that was used to evaluate the severity of colitis in previous studies, which includes the percent of body weight change, fecal viscosity, and blood stool [14].

Inflammatory cytokines involved pro-inflammatory cytokines and anti-inflammatory cytokines [15,16]. The increase in pro-inflammatory cytokines usually reflects a robustly inflammatory response from hosts. Particularly, the pro-inflammatory factors IL-1β, IL-6, and TNF-α of UC patients were significantly higher than healthy human beings [17].

The level of body inflammatory cytokines was modulated by abundant signaling pathways [16]. To further explore the concrete anti-inflammatory mechanism of SC-5, we measured the relative proteins of NF-κB and MAPK signaling pathway by Western Blotting. NF-κB is a type of important pathway that can modify the inflammatory response and participate in the response of tissue inflammation and host immunity [18]. It has been shown that the increasing inflammatory cytokines are closely related to the activation of the NF-κB signaling pathway [19,20]. At present, many anti-inflammatory drugs have been developed by inhibiting the activation of NF-κB and downregulating the level of relative proteins so that alleviate the symptom of UC patients [20]. MAPK is a group of mitogen-activated protein kinases that is responsible for delivering signals from the cell surface to the nucleus, which regulates cell growth, differentiation, and immune response [21]. Similarly, previous research has indicated that the activation of MAPK will result in the deterioration of UC [22]. Accordingly, inhibiting the level of protein expression of the MAPK signaling pathway is a significant therapeutic target.

Intestinal tight junction proteins (TJ) were considered as a type of vital protein molecule that maintains the normal function of the bowel mucosal barrier [23]. Generally, TJ proteins allow small molecules through it but prevent large molecule poison and pathogenic microorganisms from invading the intestinal barrier [24]. The TJ proteins of intestinal epithelium will be destroyed, and the integrity of the mucosal barrier will be injured when UC occurs. Consequently, a great number of harmful microorganisms and large molecules invade the intestine as a result of increased permeability in the intestine and aggravates colitis [25].

Intestinal microbiota plays a critical role in regulating gut homeostasis and modulating the immune response, which is relevant to a variety of physiological processes [26]. It has been shown that UC patients are accompanied by unbalanced intestinal microbiota that are different from common human beings [27]. More and more research has suggested that restoring the balance of gut microbiota contributes to alleviating and treating UC patients [28].

In this study, we determined the protective effect of SC-5 on DSS-induced colitis mice by assessing clinical symptoms, histopathology characteristics, pro-inflammatory cytokines, inflammatory signaling pathways, tight junction proteins, and the structure of gut microbiota. Accordingly, the SC-5 has the potential to be developed into a kind of promising agent for the prevention or treatment of UC patients in the future.

## 2. Materials and Methods

### 2.1. Materials

DSS was purchased from MP Biomedicals (Irvine, CA, USA). The *Lactiplantibacillus plantarum* subsp. *plantarum* SC-5 was provided by Jilin University. The De Man, Rogosa, and Sharpe (MRS) medium was purchased from Hopebio (Qingdao, China). Enzyme-linked immunosorbent assay (ELISA) kits for IL-1β, IL-6, and TNF-α were purchased from Biolegend (San Diego, CA, USA). The antibody for NF-κB and MAPK was purchased from Immunoway Biotechnology (Plano, TX, USA). The antibody of tight junction proteins was purchased from Wanleibio Co. (Liaoning, China). The immunofluorescence kit was purchased from Solarbio (Beijing, China).

### 2.2. Culture of Lactiplantibacillus plantarum subsp. plantarum SC-5

*Lactiplantibacillus plantarum* subsp. *plantarum* SC-5 was provided by the College of Food Science, Jilin University, and stored in a refrigerator at −80 °C. It was removed and returned to room temperature in order to expand the culture. SC-5 were cultured in De Man, Rogosa, and Sharpe (MRS) medium at speed of 180 rpm and 37 °C in a shaking incubator overnight. Then, the bacterial solution contained SC-5 were centrifuged at 10,000 rpm/min for 10 min. Phosphate buffered saline(PBS, pH = 7.4) was used to wash the collected bacteria for three times. As a result, the SC-5 resuspended in PBS was at a concentration of 1.0 × 10^9^ CFU/mL. SC-5 was cultured continuously until the end of the gavage.

### 2.3. Animals

Between 8- and 10-week-old male C57BL/6J mice (19–22 g) were provided by the Center of Experimental Animals of Jilin University and raised under specific pathogen-free (SPF) conditions (20-temperature, 24 °C; humidity, 50 ± 10% humidity; and light/dark cycle, 12 h), fed for 7 days before the experiment. All experimental procedures were conducted by the Institutional Animal Care and Use Committee of Jilin University (Nos. SY202010004).

### 2.4. DSS-Induced Colitis Mice Model and Treatment

The mice were randomly divided into three groups (Control, DSS, and SC-5). In the SC-5 group, SC-5 (1 × 10^9^ CFU per mouse per day, 0.2 mL) were orally gavaged once a day for 7 days. The DSS group and Control group were orally gavaged saline for 7 days. 2.5% DSS was added to the water during 8–14 days. The therapeutic schedule was shown in Figure 1a. Mice were sacrificed on the 15th day. Blood, colon, and stool were removed and stored at −80 °C for further experiments.

### 2.5. Evaluation of Disease Activity Index (DAI)

Body weight, stool characteristics, and rectal bleeding were recorded every day. DAI scores are calculated by previously described methods [29]. The details of marking standards were listed in Appendix A.

### 2.6. Histopathological Analysis

Colon tissue was fixed in 4% paraformaldehyde. Next, the colon samples were dehydrated with gradient alcohol and embedded in paraffin. The wax blocks were cut into slices with 5 μm. The slices were stained with hematoxylin-eosin (H&E). The details were based on method described previously [30]. According to the histopathological scores listed in Appendix A, the pathological sections were evaluated. Moreover, the slices were stained with Alcian blue and Periodic acid Schiff (AB-PAS) to observe and calculate the goblet cells in the colon crypt.

### 2.7. Measure of Inflammatory Cytokines by Enzyme-Linked Immunosorbent Assay

According to the manuscript’s protocol, the gross proteins were extracted from colon tissue, and the expression levels of IL-1β, IL-6, and TNF-α in colon tissue were determined by ELISA kit [31]. The absorbance of each sample was detected at 450 nm.

### 2.8. Western Blotting

According to the protocol described previously [32], the gross proteins were extracted from colon tissue. The proteins were separated from 10% SDS-PAGE and delivered to the PVDF membrane. The membrane was incubated with primary antibodies of p65, p-65, IκB, p-IκB, p38, p-p38, JNK, p-JNK, ERK, p-ERK, occludin, claudin-3, and GAPDH overnight at 4 °C and then with goat anti-rabbit IgG-HRP secondary antibodies for 1 h at room temperature. The membrane was visualized by an enhanced chemiluminescence (ECL) substrate and imaged. The results were quantified by ImageJ software.

### 2.9. Immunofluorescence Staining

According to the previous method [32], the colon sample was fixed in 4.0% formaldehyde and then made into slices. The slices were permeabilized with 0.25% Triton X-100. After blocking with 5.0% FBS, the tissue was incubated with the primary antibody of ZO-1, After rinsing, the slices were incubated with fluorescently labeled secondary antibodies. All images were viewed and imaged on an inverted fluorescence microscope.

### 2.10. 16S rRNA

Fresh feces from mice in the control, DSS, and SC-5 groups were sequenced by 16S rRNA sequencing through the Illumina Hiseq platform. The total DNA in feces was exacted by the Fecal genomic DNA extraction kit. The details followed the instruction. After PCR augmentation, product purification, library preparation, and quality inspection, sequencing was carried out. We used overlap to splice the RawData and performed quality control and chimera filtering to obtain high-quality CleanData. When the sequence similarity is more than 97%, it will be clustered into an operational taxonomic unit (OTU) for data analysis. According to the OUT data, we draw the diversity analysis figure in Prism8.0 to assess the alpha diversity and beta diversity of the microbial community. We analyzed the differences in community composition between each group at the phylum, class, order, genus, and species levels by Linear discriminant analysis (LDA) effect size (LEfSe). Heat maps are drawn for the correlation analysis [33].

### 2.11. Statistical Analysis

GraphPad Prism 7.0 and SPSS 23.0 were used for data analysis. A one-way analysis of variance at a significance level α = 0.05 was used to assess significant differences among groups.

## 3. Results

### 3.1. Lactiplantibacillus plantarum subsp. plantarum SC-5 Alleviates Clinical Symptoms of DSS-Induced Colitis Mice

The effect of SC-5 on DSS-induced colitis was evaluated in C57BL/6J mice. Compared with the Control group, the body weight of the DSS group showed a significant decrease (*p* < 0.05). After treatment with SC-5, compared with the DSS group, the weight loss of mice in the SC-5 group decreased (Figure 1b). As shown in Figure 1c, the length of the colon of mice treated with DSS was significantly shorter than that of the control group, while the treatment of SC-5 significantly increased the length of the colon of mice (*p* < 0.05), which was closer to that of the control group (Figure 1d). The DAI score representing the severity of colitis also decreased in the SC-5 group (Figure 1e). These results suggested that SC-5 can alleviate the clinical symptoms of colitis in mice.

### 3.2. Lactiplantibacillus plantarum subsp. plantarum SC-5 Improves the Histopathological Damage of Colonic Tissues

The H&E staining of colon tissue is shown in Figure 2a. As the figure depicts, the mice in the control group had a more complete intestinal mucosal epithelial structure, with intact glands, no inflammatory cell infiltration, and no exfoliated intestinal epithelial cells in the gut. After DSS treatment, the colonic crypts of mice were severely damaged by the DSS with large amounts of exfoliated intestinal epithelial cells. Under the function of DSS, the structure of lamina propria, submucosa, muscular, and serosa was significantly damaged. The edge of the intestinal was blurred and the large intestinal glands almost disappeared, with a large number of inflammatory cells infiltrating in the intestinal mucosal epithelium, such as neutrophils, lymphocytes, and basophils. These results showed extremely severe histopathological damage in the gut. However, the treatment with SC-5 significantly improved this situation. As shown in the H&E staining of the SC-5 treatment group, we found that the structure of the intestinal mucosal epithelium in the colon of mice tended to be complete, and large intestinal glands were regenerated. The boundaries between lamina propria, submucosa, muscular, and serosa were clear, and inflammatory cells were reduced, which was close to the control group. The histopathological score was significantly lower than that of the DSS group (*p* < 0.05), which indicated that SC-5 could protect the colon tissue from the damage of DSS treatment (Figure 2b).

### 3.3. Lactiplantibacillus plantarum subsp. plantarum SC-5 Promotes Goblet Cell Regeneration and Secretion of Acidic Mucin

AB-PAS staining was shown in Figure 2c. The dye results will show blue when the staining solution combines with acidic mucin, red when combined with neutral mucin, and blue–purple when combined with mixed mucin. According to this rule, we observed that a large number of goblet cells in the colon tissue of the control group were blue and blue–purple, which indicated the acidic atmosphere of the colon in general. Moreover, the intestinal mucus layer was thicker than other groups. Compared with the control group, the thickness of the intestinal mucus layer in mice treated with DSS was significantly reduced, goblet cells were lost in large quantities, and the level of acidic mucin was significantly reduced, indicating the normal acid-base conditions of colon tissue was changed by DSS. The colonic tissue treated with SC-5 is shown in Figure 2c. We found that SC-5 can significantly restore the normal acid-base conditions of the colonic tissue, increase the thickness of the mucus layer of colon tissue, and the number of goblet cells, and restore the damaged intestinal mucosal barrier (*p* < 0.05) (Figure 2d).

### 3.4. Lactiplantibacillus plantarum subsp. plantarum SC-5 Reduces Levels of Proinflammatory Cytokines

IL-1β, IL-6, and TNF-α of colon tissue were determined by ELISA. Compared with the control group, the level of pro-inflammatory cytokines increased significantly in the DSS group (*p* < 0.05) (Figure 1f–h). IL-1β, IL-6, and TNF-α were almost threefold higher than the control group, respectively. After treatment with SC-5, the level of pro-inflammatory cytokines in the colon tissue of mice was significantly lower than that in DSS treated group (*p* < 0.05). IL-1β and TNF-α had a certain decline. Moreover, the decline of IL-6 was more significant, which was closer to the level of the control group. Therefore, SC-5 can significantly reduce the inflammatory cytokines of the colon tissue in DSS-induced colitis mice, indicating that it had a good anti-inflammatory effect.

### 3.5. Lactiplantibacillus plantarum subsp. plantarum SC-5 Ameliorates DSS-Induced Colitis in Mice by Inhibiting the Activation of NF-κB and MAPK Signaling Pathways

Inflammation is closely related to the activation of NF-κB and MAPK signaling pathways. According to the results of the Western blot shown in (Figure 3a,b), we found that SC-5 could inhibit the activation of NF-κB by down-regulating the expression of p-IκB and p-p65. Similarly, SC-5 treatment can also down-regulate the expression of related proteins mediated by the MAPK signaling pathway, such as p-p38, p-ERK, and p-JNK. The results demonstrated that SC-5 could inhibit the activation of NF-κB and MAPK to improve the inflammatory response of the colon induced by DSS in mice.

### 3.6. Lactiplantibacillus plantarum subsp. plantarum SC-5 Alleviates DSS-Induced Colitis in Mice by Enhancing Tight Junction Protein Expression

As Figure 4a,b depict, the expression of tight junction proteins occludin and claudin-3 were significantly down-regulated in the colon tissue of mice treated with DSS (*p* < 0.05). According to the immunofluorescence staining shown in Figure 4c, it was found that in the immunofluorescence section of the colon tissue of the DSS group, the staining sites in the tissue that stained with ZO-1 were less than those in the control group. It was shown that the expression of ZO-1 was also down-regulated in DSS-induced murine colitis tissue. After treatment with SC-5, the expression levels of occludin, claudin-3, and ZO-1 were up-regulated compared with the DSS group (*p* < 0.05), indicating that SC-5 can alleviate DSS-induced colitis in mice by enhancing the expression of tight junction proteins.

### 3.7. Lactiplantibacillus plantarum subsp. plantarum SC-5 Alleviates DSS-Induced Colitis in Mice by Altering Intestinal Microbiota

According to the results of 16S rRNA, we used Chao1 and Simpson to describe the alpha diversity. We found that DSS treatment significantly increased community richness compared with the control group (*p* < 0.05). DSS and SC-5 group had no significant difference (*p >* 0.05) (Figure 5a). Moreover, DSS significantly increased community diversity (*p* < 0.05). However, SC-5 treatment restored the community diversity to an extent (Figure 5b). The beta diversity was observed by NMDS analysis in Figure 5c. We also found that beta diversity was restored by SC-5, which was similar to the control group. These results show that SC-5 can restore the richness, diversity, and uniformity of the communities and make them closer to the control group.

As shown in Figure 5e, at the phylum level, DSS treatment significantly decreased the abundance of *Bacteroidetes* within the mice intestinal microbiota, whereas treatment with SC-5 significantly increased the relative abundance of *Bacteroidetes* as well as *Desulfobacterota*, *Campylobacterota*, *Verrucomicrobiota*, and *Proteobacteria* (*p* < 0.05). Moreover, the SC-5 treatment significantly decreased the proportion of *Firmicutes*/*Bacteroidetes* (*p* < 0.05) (Figure 5d). At the level of genus, SC-5 reduced the abundance of the *Helicobacter* and *Desulfovibrionaceae*, and increased *Clostridiales*, *Eubacterium* and *Lachnospiraceae* compared with the DSS group (Figure 5f). These results demonstrated the change of microbiota composition caused by DSS treatment was restored to a certain extent through SC-5 treatment.

We used correlation analysis to determine the relationship between the gut microbiota and the length of the colon, DAI score, histopathology score, inflammatory cytokines, and tight junction proteins (Figure 6a). *Lachnospiraceae*, *Lactobacillus*, and *Muribaculaceae* were significantly and positively correlated with the length of the colon, and tight junction proteins occludin and claudin-3 (*p* < 0.05). There was a significant negative correlation between *Lachnospiraceae*, *Lactobacillus*, and *Muribaculaceae* and DAI score, histopathology score, and inflammatory cytokines (*p* < 0.05). Moreover, *Desulfovibrio* and *Escherichia* also had a significantly positive correlation with pro-inflammatory cytokines and a significantly negative correlation with the length of the colon, and tight junction proteins occludin and claudin-3 (*p* < 0.05).

The results of LEfSe analysis were shown in Figure 6b, we found that 68 key phylotypes as distinguished biomarkers were determined on the genus. 10 dominant microorganisms were found in the control group, 13 in the DSS group, and 8 in the SC-5 group (Figure 6c).

## 4. Discussion

The pathogenesis of IBD is closely associated with many factors, such as environmental factors, genetic factors, immune balance, and intestinal microbial conditions [34]. The treatment of IBD is mainly aimed at improving and alleviating the symptoms of patients. Although the long-term use of antibiotics and immunomodulators can alleviate the symptoms of the disease, it also increases the risk of other complications, with a high recurrence rate of IBD [35]. However, the application of *Lactiplantibacillus plantarum* subsp. *plantarum* can significantly reduce the recurrence rate of IBD patients without the above side effects [36]. In this study, SC-5 significantly ameliorated DSS-induced weight loss, colon shortening, diarrhea, and bloody stools in mice.

Clinical symptoms are the macroscopic manifestations of the microscopic changes in the body. On this basis, we observed the H&E staining and AB-PAS staining sections of the colon tissue of colitis mice treated with SC-5 and found that SC-5 could indeed alleviate the histopathological damage of colitis in mice. SC-5 treatment can significantly reduce the infiltration of inflammatory cells and restore the disordered structure of colonic crypts and large intestinal glands. Promote the regeneration of goblet cells and the secretion of acidic mucin. In previous studies, it was found that the crypt base and muscular mucosa of the colon tissue were often separated loosely in mice with DSS-induced colitis [37], which was also verified in this study. The treatment of SC-5 significantly improved this situation, which strengthened the tightness between the intestinal mucosal structures. The integrity of a good physical structure is fundamental, and the intestinal barrier plays a protective role [38]. SC-5 improved the integrity of the intestinal mucosal barrier and decreased intestinal permeability, which had a significant protective effect on DSS-induced colitis in mice.

The increase in pro-inflammatory cytokines is an important manifestation that the organisms resist inflammation. The pro-inflammatory cytokines in the serum and colon tissue of IBD patients usually increase significantly [39]. We verified this theory by ELISA and found that IL-1β, IL-6, and TNF-α in the colon tissue of DSS-induced colitis mice were significantly increased, showing a strong anti-inflammatory process. However, the levels of IL-1β, IL-6, and TNF-α in the colon tissue of mice treated with SC-5 showed a downward trend, indicating that the inflammatory response of the body was alleviated. To explore the specific anti-inflammatory mechanism of SC-5, we used Western Blot to measure the proteins expression levels of NF-κB and MAPK signaling pathway and found that SC-5 inhibited the activation of NF-κB pathway by down-regulated the expression level of p-p65, p-IκB. Similarly, SC-5 also inhibited the activation of the MAPK signaling pathway by down-regulates p-p38, p-JNK, and p-ERK [40], which explained how SC-5 plays an anti-inflammatory role in the body.

The intestinal epithelial barrier is mainly maintained by intestinal epithelial tight junction proteins [41]. Tight junction proteins are a class of complex supramolecular entities that often participate in complex protein-protein interactions [42]. Through Western blot and immunofluorescence, we found that DSS treatment severely damaged the structure of tight junction protein and activated related inflammatory pathways, which was consistent with the DSS mouse colitis model established by Sharma, D. [43]. The decrease in tight junction protein destroys the integrity of intestinal mucosal barrier, which seriously destroys the isolation mechanism between intestinal lumen contents and intestinal epithelial barrier, thus making macromolecular immunogenic substances more likely to pass through the highly permeable intestinal mucosa and induce abnormal immune activation [44]. As a result, the incidence of IBD and various gastrointestinal diseases has increased. SC-5 treatment restored the integrity of the intestinal mucosal barrier by enhancing the expression of intestinal epithelial tight junction protein ZO-1, occludin, and claudin-3, and reduced the adverse interaction between intestinal contents and intestinal mucosal barrier. It plays a vital role in alleviating the severity of inflammation in the DSS-induced colitis of mice.

Recent research has found that intestinal microbiota is closely related to a variety of physiological and pathological processes of organisms [45]. The balance of intestinal microbiota is the basis for the body to maintain normal physiological functions [46,47]. The results of 16S rRNA in the DSS group were similar to the changes in intestinal microbiota in IBD patients [48]. Generally, the intestinal microbiota of healthy people is mainly composed of *Firmicutes* and *Bacteroidetes* [45], but the microbiota structure of IBD patients is significantly destroyed [49,50,51,52,53]. In this study, DSS treatment significantly increased *Firmicutes*/*Bacteroidetes*, which greatly changed the composition of the normal intestinal microbiota and destroyed the balance of intestinal microbiota. Moreover, the abundance of harmful microbes increased by DSS treatment, such as *Desulfobacterota*, and *Proteobacteria*, and has been verified to be harmful to intestinal homeostasis [54,55,56]. Interestingly, SC-5 treatment significantly decreased the abundance of harmful microbes and increased the richness of *Clostridiales*, *Eubacterium*, and *Lachnospiraceae. Clostridiales* have a potent anti-tumor effect, which can inhibit the development of colorectal cancer [57]. *Eubacterium* plays a critical role in colonic motility, immunomodulation, and suppression of inflammation in the intestine [58]. Previous studies have shown that the decreased abundance of *Lachnospiraceae* may trigger the recurrence of UC [59], but SC-5 significantly reversed this situation. SC-5 treatment increased the abundance of beneficial bacteria in the gut, making the microbial composition of DSS-induced colitis mice closer to the control group, which is consistent with the study of Hao [60]. Accordingly, *Lactiplantibacillus plantarum* subsp. *plantarum* SC-5 can significantly restore the richness, diversity, and uniformity of community in the gut. It is beneficial to remodel intestinal microbial community composition and maintain intestinal homeostasis [61].

According to the correlation analysis, we found that *Lachnospiraceae*, *Lactobacillus*, and *Muribaculaceae* have negative correlations with pro-inflammatory cytokines, which showed good anti-inflammatory characteristic in DSS-induced colitis in mice. Indeed, studies have shown that *Muribaculaceae* plays an anti-inflammatory role in the intestine and contributes to alleviating UC [60]. However, *Desulfovibrio* and *Escherichia* may promote the production of pro-inflammatory cytokines and exacerbate colitis’s progression. In a previous study, it has been verified that *Desulfovibrio* can produce sulfide which plays a vital role in bowel inflammation, including UC [62]. Therefore, it also provides new insight into the prognosis of IBD patients.

In conclusion, SC-5 can protect mice from DSS-induced colitis by inhibiting the activation of NF-κB and MAPK signaling pathways, strengthening tight junction proteins, improving the integrity of the intestinal mucosal barrier, and restoring the balance of gut microbiota structure. It is a probiotic candidate with good prospects and potential and is expected to be developed into a novel probiotic medicine for preventing and treating IBD.

## Figures and Tables

**Figure 1 foods-12-00897-f001:**
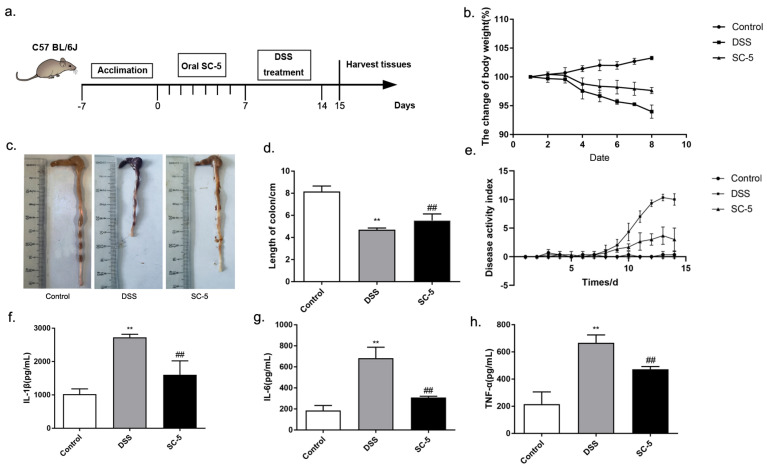
Effects of SC-5 supplementation on clinical symptoms of DSS-induced colitis mice and the level of pro-inflammatory cytokines. (**a**) Experimental procedure; (**b**) Weight changes; (**c**) colon; (**d**) colon length; (**e**) DAI score; (**f**) the level of IL-1β; (**g**) IL-6; (**h**) TNF-α. All data are expressed as mean ± SD (*n* = 8). ** *p* < 0.01 vs. the control group; ## *p* < 0.01 vs. the DSS group.

**Figure 2 foods-12-00897-f002:**
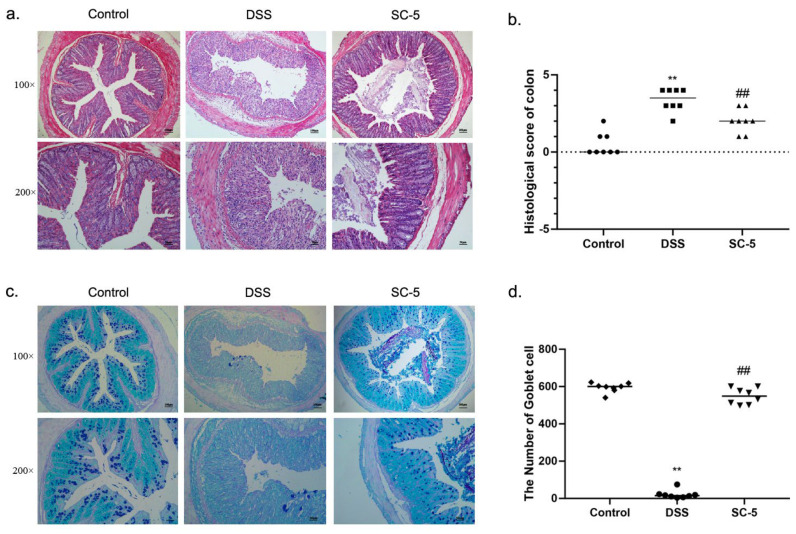
Effects of SC-5 supplementation on colonic histological evaluation. (**a**) H&E strain; (**b**) histopathology scores; (**c**) AB-PAS strain; (**d**) the number of goblet cells; All data are expressed as mean ± SD (*n* = 8). ** *p* < 0.01 vs. the control group; ## *p* < 0.01 vs. the DSS group.

**Figure 3 foods-12-00897-f003:**
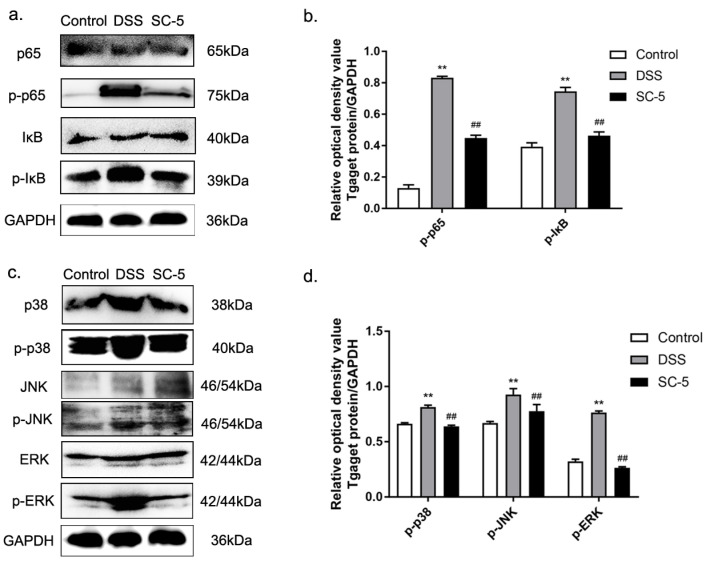
Effects of SC-5 supplementation on the levels of relative proteins expression in NF-κB and MAPK pathways. (**a**) NK-κB p65 and IκB phosphorylation levels were analyzed by Western blotting. (**b**) The relative expression of NF-κB p65 and IκB phosphorylation levels was normalized to GAPDH; (**c**) MAPK p38, JNK, and ERK phosphorylation levels were analyzed by Western blotting; (**d**) The relative expression of MAPK p38, JNK, and ERK phosphorylation levels were normalized to GAPDH. All data are expressed as mean ± SD (*n* = 8). ** *p* < 0.01 vs. the control group; ## *p* < 0.01 vs. the DSS group.

**Figure 4 foods-12-00897-f004:**
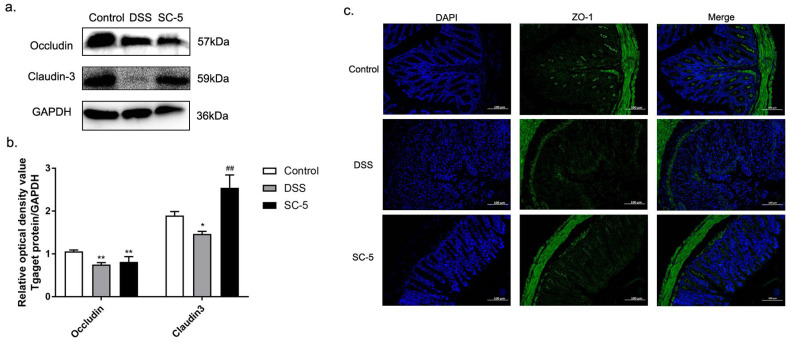
Effects of SC-5 supplementation on the level of tight junction proteins. (**a**) Levels of occludin, claudin-3; (**b**) The relative expression of occludin, claudin-3 was normalized to GAPDH. (**c**) Representative images of control, DSS, and SC-5 at the level of ZO-1 by immunofluorescence staining. The image was taken by a confocal microscope with ×200 magnification. Blue: DAPI in the nucleus, green: ZO-1 in the colon. All data are expressed as mean ± SD (*n* = 8). * *p* < 0.05 and ** *p* < 0.01 vs. the control group; ## *p* < 0.01 vs. the DSS group.

**Figure 5 foods-12-00897-f005:**
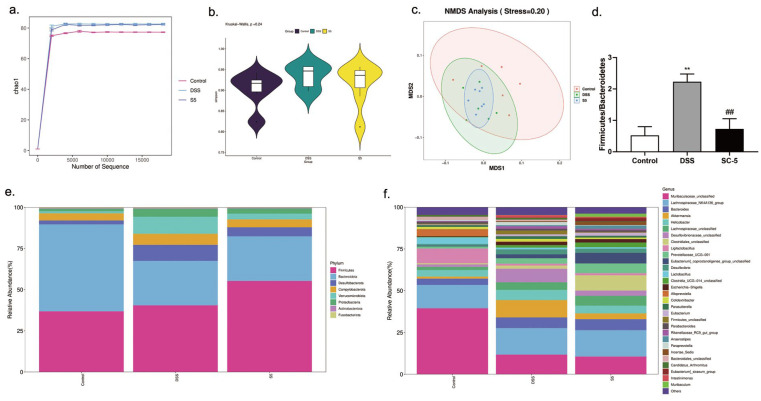
Effects of SC-5 supplementation on the diversity and composition of gut microbiota. (**a**) Chao1; (**b**) Simpson; (**c**) NMDS Analysis; (**d**) *Firmicutes/Bacteroidetes*; (**d**) phylum level; (**e**) genus level; All data are expressed as mean ± SD (*n* = 3). ** *p* < 0.01 vs. the control group; ## *p* < 0.01 vs. the DSS group.

**Figure 6 foods-12-00897-f006:**
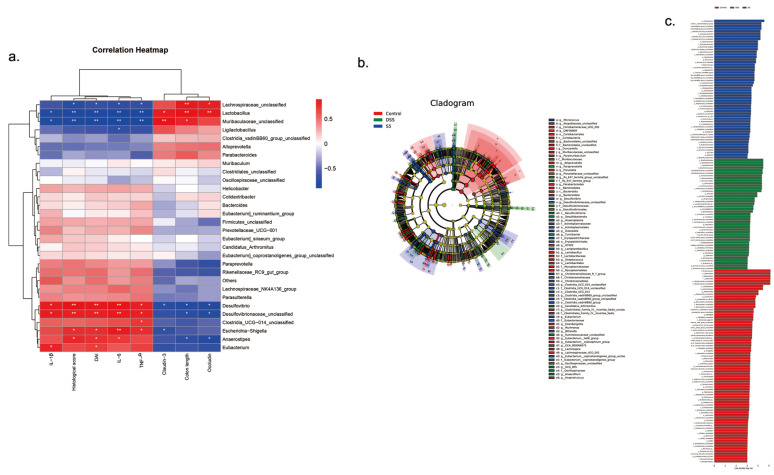
The effect of SC-5 on gut dominant microorganisms and its correlation analysis (**a**) Correlation between gut microbiota and colon length, DAI score, histopathology score, IL-β, IL-6, TNF-α, occludin, and claudin-3; (**b**) Cladogram displayed the taxonomic tree of differentially abundant taxa by LEfSe analysis on the genus. (**c**) Distribution histogram of gut dominant microorganisms on the genus. All data are expressed as mean ± SD (*n* = 3).

## Data Availability

Data is contained within the article or Appendix A.

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
