# Peer review of "Protective Effect of Lactiplantibacillus plantarum subsp. plantarum SC-5 on Dextran Sulfate Sodium—Induced Colitis in Mice"

_foods, 2023, doi:10.3390/foods12040897_

Round 1

Reviewer 1 Report

In this study, Shi et al investigate whether L. plantarum SC-5 is able to mitigate DSS-induced colitis in mice. The authors find a substantial reduction in disease severity for the SC-5-treated group, and measure other aspects of disease severity, including gene expression and microbiome composition. In general, I find this study to be straightforward.  However, I recommend a few clarifying points:

1.       Titles similar to “SC-5 ameliorates colitis by ….” are a bit misleading.  The phenomena the authors observe are correlates of SC-5 administration, and it is unclear whether these phenomena (e.g. microbiome modulation) actually cause improvement in symptoms. More care should be given in discussing and interpreting these results.

2.       The authors should discuss how unique SC-5 is in its ability to ameliorate DSS-induced colitis relative to other strains of Lactobacillus.  For example, does other literature show that other Lactobacillus strains are effective too?  If so, how does SC-5 compare?

3.       There is a lack of some details on how SC-5 was administered.  It is necessary to know the buffer SC-5 was suspended in, how cultured SC-5 was prepared for gavage, etc.

4.       I recommend a detailed check for grammar and flow, as there were several instances of typos and awkward phrasing.

Author Response

Response to Reviewer 1 Comments

Thank you for your letter and the reviewers’ comments concerning our manuscript entitled “Protective effect of Lactiplantibacillus plantarum subsp. plantarum SC-5 on dextran sulfate sodium-induced colitis in mice”. These comments are all valuable and very helpful for revising and improving our paper, as well as the important guiding significance to our researches. Below are our responses to the editors' and reviewers' comments.

Point 1: Titles similar to “SC-5 ameliorates colitis by ….” are a bit misleading. The phenomena the authors observe are correlates of SC-5 administration, and it is unclear whether these phenomena (e.g. microbiome modulation) actually cause improvement in symptoms. More care should be given in discussing and interpreting these results.

Response 1: We have modified the discussion more carefully. Thank you for your suggestions.

Point 2: The authors should discuss how unique SC-5 is in its ability to ameliorate DSS-induced colitis relative to other strains of Lactobacillus. For example, does other literature show that other Lactobacillus strains are effective too? If so, how does SC-5 compare?

Response 2: We have added other references to Lactiplantibacillus plantarum subsp. plantarum for the treatment of colitis in mice. Such as Lactiplantibacillus plantarum subsp. plantarum LP-Onlly, Lactobacillus reuteri S5, Lactiplantibacillus plantarum subsp. plantarum L15 and so on. These strains have similar effects to Lactiplantibacillus plantarum subsp. plantarum SC-5. Thanks for your suggestion.

Point 3: There is a lack of some details on how SC-5 was administered. It is necessary to know the buffer SC-5 was suspended in, how cultured SC-5 was prepared for gavage, etc.

Response 3: We have added the details about Lactiplantibacillus plantarum subsp. plantarum SC-5 in subchapters p 2.2. Thanks for your suggestion.

Point 4: I recommend a detailed check for grammar and flow, as there were several instances of typos and awkward phrasing.

Response 4: We have checked and corrected the grammar and some instances of the manuscript. Thanks for your suggestion.

Reviewer 2 Report

This is an impressive work identifying the potential of L. plantarum SC-5 to treat IBD-related diseases. Some minor changes is needed.

In section 2.10, please rewrite 16srRNA.

Please use the new name of Lactobacillus plantarum throughout the manuscript as it has been changed since April 2020.

Author Response

Response to Reviewer 2 Comments

Thank you for your letter and the reviewers’ comments concerning our manuscript entitled “Protective effect of Lactiplantibacillus plantarum subsp. plantarum SC-5 on dextran sulfate sodium-induced colitis in mice”. These comments are all valuable and very helpful for revising and improving our paper, as well as the important guiding significance to our researches. Below are our responses to the reviewers' comments.

Point 1: In section 2.10, please rewrite 16srRNA.

Response 1: We have rewritten the related content of 16SrRNA. Thank you.

Point 2: Please use the new name of Lactobacillus plantarum throughout the manuscript as it has been changed since April 2020.

Response 2: We have used the new name” Lactiplantibacillus plantarum subsp. plantarum SC-5” instead of Lactobacillus plantarum SC-5. Thank you.

Reviewer 3 Report

It is an interesting contribution to research in probiotics. However, some corrections are necessary.

Title and in all the document. Use Lactiplantibacillus plantarum subsp. plantarum instead of Lactobacillus plantarum.

General. Please insert line numbers in the document to make easier the reviewing process. I am including a copy of the document with the corrections highlighted in yellow.

p 1, 2 and in the whole document. Please use "microbiota" instead of "flora"

p 2. Use "assess" instead of "access". Check the whole manuscript.

p 2. Please mention at least 3 probiotic microorganisms that have been used previously to treat and prevent gastrointestinal diseases.

p 3 and 4. References are needed for subchapters 2.6, 2.7, 2.8, 2.9 and 2.10. 

p 4. In subchapter 2.11, the correct sentence is "A one-way analysis of variance at a significance level α=0.05 was used to assess significant differences among groups.". Delete the "Differences were considered..." sentence.

p 6. The authors indicate that SC-5 reduced the abundance of Akkermancia. This change in microbiota is not a good signal since it is usually an indication of dysbiosis. Akkermancia is supposed to be abundant in good health conditions. Please explain. 

Author Response

Response to Reviewer 3 Comments

Thank you for your letter and the reviewers’ comments concerning our manuscript entitled “Protective effect of Lactiplantibacillus plantarum subsp. plantarum SC-5 on dextran sulfate sodium-induced colitis in mice”. These comments are all valuable and very helpful for revising and improving our paper, as well as the important guiding significance to our researches. Below are our responses to the  reviewers' comments.

Point 1:p 1, 2, and in the whole document. Please use "microbiota" instead of "flora"

Response 1: We have used "microbiota" instead of "flora". Thanks for your suggestion.

Point 2:p 2. Use "assess" instead of "access". Check the whole manuscript.

Response 2: We have used "assess" instead of "access". Thanks for your suggestion.

Point 3:p 2. Please mention at least 3 probiotic microorganisms that have been used previously to treat and prevent gastrointestinal diseases.

Response 3: We have mentioned 3 probiotic microorganisms in the introduction. Such as Lactiplantibacillus plantarum subsp. plantarum LP-Onlly, Lactobacillus reuteri S5, Lactiplantibacillus plantarum subsp. plantarum L15 and so on. These probiotic microorganisms have a good effect to treat and prevent gastrointestinal diseases. Thanks for your suggestion.

Point 4:p 3 and 4. References are needed for subchapters 2.6, 2.7, 2.8, 2.9, and 2.10.

Response 4: References have been added to the manuscript in subchapters 2.6, 2.7, 2.8, 2.9, and 2.10. Thanks for your suggestion.

Point 5:p 4. In subchapter 2.11, the correct sentence is "A one-way analysis of variance at a significance level α=0.05 was used to assess significant differences among groups.". Delete the "Differences were considered..." sentence.

Response 5: We have corrected the sentence in subchapter 2.11, and deleted the "Differences were considered..." sentence. Thanks for your suggestion.

Point 6: p 6. The authors indicate that SC-5 reduced the abundance of Akkermancia. This change in microbiota is not a good signal since it is usually an indication of dysbiosis. Akkermancia is supposed to be abundant in good health conditions. Please explain.

Response 6: According to the 16SrRNA results, we found that the abundance of Akkermancia in the intestinal flora of DSS-treated mice was significantly increased. Recent studies have shown that Akkermancia is indeed a probiotic beneficial to intestinal homeostasis, but DSS treatment increased the amount of Akkermancia. Our experimental results are inconsistent with expectations.

Based on this, through a large number of references, we found that the proportion of Akkermancia in intestinal flora was significantly associated with animal food intake.

According to the study of Sonoyama, K. et al [1-3], fasting treatment will lead to a significant increase in the abundance of Akkermancia in the intestine. It was also found that the proportion of Akkermancia in the intestine of colon cancer patients with reduced food intake was significantly lower than that of healthy adults.[4].

It may be due to the dominant position of Akkermancia in the competition with other flora when the animal's food intake was reduced and malnutrition occurred.

Interestingly, in the course of the experiment, we observed that the mice treated with DSS had severe intestinal inflammation, obvious symptoms of diarrhea and hematochezia, and a significant reduction in food intake and water intake.

Therefore, we speculated that Akkermancia had a competitive advantage in the intestinal flora at this time, and the proportion of Akkermancia in the intestinal flora of mice in the DSS group was significantly increased. Thanks for your suggestion.

Reference:

  1. Sonoyama, K., et al., Response of gut microbiota to fasting and hibernation in Syrian hamsters. Appl Environ Microbiol, 2009. 75(20): p. 6451-6.
  2. Remely, M., et al., Increased gut microbiota diversity and abundance of Faecalibacterium prausnitzii and Akkermansia after fasting: a pilot study. Wien Klin Wochenschr, 2015. 127(9-10): p. 394-8.
  3. Remely, M., et al., Gut microbiota composition correlates with changes in body fat content due to weight loss. Benef Microbes, 2015. 6(4): p. 431-9.
  4. Weir, T.L., et al., Stool microbiome and metabolome differences between colorectal cancer patients and healthy adults. PLoS One, 2013. 8(8): p. e70803.
